# Comprehensive characterization of claudin-low breast tumors reflects the impact of the cell-of-origin on cancer evolution

Roxane M. Pommier [1,2,3], Amélien Sanlaville [1,2], Laurie Tonon[3], Janice Kielbassa[3], Emilie Thomas[3], Anthony Ferrari[3], Anne-Sophie Sertier[3], Frédéric Hollande [4], Pierre Martinez[1,2], Agnès Tissier [1,2], Anne-Pierre Morel[1,2], Maria Ouzounova [1,2] & Alain Puisieux [1,2,5✉]

Claudin-low breast cancers are aggressive tumors defined by the low expression of key components of cellular junctions, associated with mesenchymal and stemness features. Although they are generally considered as the most primitive breast malignancies, their histogenesis remains elusive. Here we show that this molecular subtype of breast cancers exhibits a significant diversity, comprising three main subgroups that emerge from unique evolutionary processes. Genetic, gene methylation and gene expression analyses reveal that two of the subgroups relate, respectively, to luminal breast cancers and basal-like breast cancers through the activation of an EMT process over the course of tumor progression. The third subgroup is closely related to normal human mammary stem cells. This unique subgroup of breast cancers shows a paucity of genomic aberrations and a low frequency of *TP53* mutations, supporting the emerging notion that the intrinsic properties of the cell-of-origin constitute a major determinant of the genetic history of tumorigenesis.

[1] Cancer Research Center of Lyon, Université de Lyon, Université Claude Bernard Lyon 1, INSERM 1052, CNRS 5286, Centre Léon Bérard, Equipe Labellisée Ligue contre le Cancer, 69008 Lyon, France. [2] LabEx DEVweCAN, Université de Lyon, F-69000 Lyon, France. [3] Fondation Synergie Lyon Cancer, Plateforme de bioinformatique Gilles Thomas, Centre Léon Bérard, Lyon, France. [4] Department of Clinical Pathology, The University of Melbourne, Victorian Comprehensive Cancer Centre, Parkville, Victoria, Australia. [5] Institut Curie, PSL Research University, Paris, France. ✉email: alain.puisieux@curie.fr

**B**reast cancer is a highly heterogeneous group of diseases with variable biological and clinical behaviors. Providing early insights into this diversity, gene-expression profiling analyses initially resulted in the identification of four clinically relevant molecular subtypes, known as intrinsic subtypes (luminal A, luminal B, HER2-enriched and basal-like, according to the PAM50 classification) mostly corresponding to hormone receptor and HER2 status, and a normal breast-like group[1–4]. Among these intrinsic subtypes, the basal-like subtype appears to be the most distinct, as it is characterized by the unique expression of cytokeratins typically expressed by the basal layer of the skin and a very low level of expression of luminal-related genes[2,5]. This observation led to the hypothesis that breast cancers may arise from the transformation of two distinct cell types of origin or developmental stages of mammary epithelial cell development, one generating basal-like tumors and the other non-basal-like malignancies[5]. Although appealing in its simplicity, this model overlooks the potential reprogramming of lineage-restricted populations initiated by an oncogenic event or a microenvironmental signal over the course of tumor development[6–8]. Moreover, it does not account for the intrinsic diversity of each breast cancer subtype, notably basal-like tumors known for their great heterogeneity[9–11]. An additional intrinsic subtype of breast cancers, known as claudin-low, has recently been identified in human and mouse tumors and in breast cancer cell lines, showing several common features with basal-like tumors and reflecting the diversity of tumors with a low luminal differentiation status[4,12]. Basal-like and claudin-low tumors form the majority of triple-negative breast cancers (TNBCs), an aggressive subgroup of breast malignancies defined as tumors lacking expression of the estrogen receptor (ER), progesterone receptor (PR), and HER2. A hallmark of the claudin-low subtype is the low expression level of critical cell–cell adhesion molecules, including claudins 3, 4, and 7, occludin, and E-cadherin. Tumors of this subtype are highly enriched in mesenchymal traits and stem cell features and are therefore considered as the most primitive breast cancers[13]. Although in vivo preclinical data suggested that basal-like tumors arise from the transformation of a luminal progenitor (LP)[14], the putative cell-of-origin of claudin-low tumors remains unknown. The prevailing hypothesis is that, over the course of tumor progression, basal cancer cells undergo an epithelial-mesenchymal transition (EMT) in response to acquired oncogenic events and/or microenvironmental signals, thereby gaining mesenchymal and stemness features[15–17]. An alternative hypothesis is that claudin-low tumors originate from an early epithelial precursor with inherent stemness features[4,13]. To gain further insight into the developmental origin of claudin-low tumors, we first sought to analyze their genomic architecture. Indeed, we have recently demonstrated that, whereas the oncogene-driven transformation of mature luminal (mL) cells and progenitor luminal cells triggers massive oncogene-induced DNA damage and an early onset of chromosomal instability (CIN), normal human mammary stem cells (MaSCs) can withstand an aberrant mitogenic activity[18]. The endogenous expression of the ZEB1 EMT-inducing transcription factor prevents replication and oxidative stress, leading to a process of malignant transformation in the absence of exacerbated genomic instability[18]. As basal-like breast cancers generally exhibit numerous genomic aberrations, we thus speculated that the extent of genomic aberrations might be used as a molecular indicator of the developmental origin of claudin-low breast cancers.

To comprehensively characterize the claudin-low breast tumor subtype, we used a multi-omics approach that reveal the existence of three distinct subgroups with specific transcriptomic, epigenetic, and genetic characteristics, strongly supporting the hypothesis of different cells-of-origin.

## Results

**Selection of claudin-low tumors.** Claudin-low tumors exhibit marked immune and stromal cell infiltration, when compared with all other breast cancer subtypes[4,19]. As gene expression analyses do not allow to accurately discriminate mesenchymal cancer cells from normal stromal cells, we used the allele-specific copy number analysis of tumors (ASCAT) copy number-based tumor purity estimation method to assess tumor cell fraction[20] (Supplementary Fig. 1a). To avoid any substantial bias due to non-tumor cell contamination, a stringent purity threshold was determined by applying Wilcoxon test, for which the degree of contamination of claudin-low tumors was not statistically different from that of non-claudin-low tumors, thus enabling a comparable distribution of tumors (Supplementary Fig. 1b). Using this stringent threshold, 45 out of 152 tumors classified as claudin-low from Molecular Taxonomy of Breast Cancer International Consortium (METABRIC) were selected for further studies. Demonstrating an unexpected degree of diversity, only 35.6% of these tumors were classified as TNBCs (Supplementary Fig. 1d). This result was not due to the purity selection, as the percentage of TNBCs within all claudin-low tumors was similar before applying the purity threshold (Supplementary Fig. 1c). Of note, only four normal-like tumors were estimated as pure. Due to the lack of statistical power, they were excluded from subsequent analyses.

**Identification of CNA-devoid claudin-low tumors.** We next analyzed claudin-low tumors using the stratification based on gene expression and copy number alterations (CNAs), previously defined by Curtis et al.[21,22] (Fig. 1a). As expected for aggressive neoplasms, basal-like tumors mostly belonged to the integrative cluster 10, characterized by multiple CNAs affecting most chromosomes. Luminal A and B tumors were distributed across clusters 1, 2, 3, 6, 7, 8, and 9, whereas HER2-positive tumors were mainly found in cluster 5. Interestingly, claudin-low tumors were stratified into three main clusters, again demonstrating a significant heterogeneity. Only 17.8% were found in cluster 10, whereas 48.8% and 17.8% were found in integrative clusters 4 and 3, respectively, regardless of tumor purity when compared with non-claudin-low tumors (Supplementary Fig. 2). Both integrative clusters 3 and 4 are marked by a paucity of copy number and cis-acting alterations. Integrative cluster 3 displays a simplex pattern of rearrangements, dominated by whole arm gain of chromosome 1q and 16p and loss of 16q[11,21,22]. It was initially described as essentially composed of luminal breast tumors. Integrative cluster 4 includes both ER-positive and ER-negative cases. It was defined as a CNA-devoid subgroup, with an expression profile dominated by immune-related genes, leading to the hypothesis that it may be essentially composed of samples with a high infiltration of normal cells[11]. As the method used in the present study to select tumors with a high index of purity does not rely on gene expression patterns, it argues against this hypothesis. Nevertheless, to formally demonstrate the existence of claudin-low tumors with a flat genomic landscape, we analyzed the genomic architecture of tumor samples exhibiting a very low level of aberrations (fraction of genome altered, FGA < 1%). After having previously removed germline copy number variations (CNVs), we analyzed B allele frequency (BAF) and log R ratio (LRR) profiles of single-nucleotide polymorphisms (SNPs) localized in genomic regions of deletion (LRR segment mean cutoff value < −0.4) (Fig. 1b and Supplementary Fig. 3a, c). This analysis led to the identification of regions of single-copy allelic loss displaying a clear separation between BAF values (<0.8 and >0.2). As previously demonstrated, this observation is incompatible with a massive contamination of the tumor tissue by normal cells[23]. Consistent with these data, the

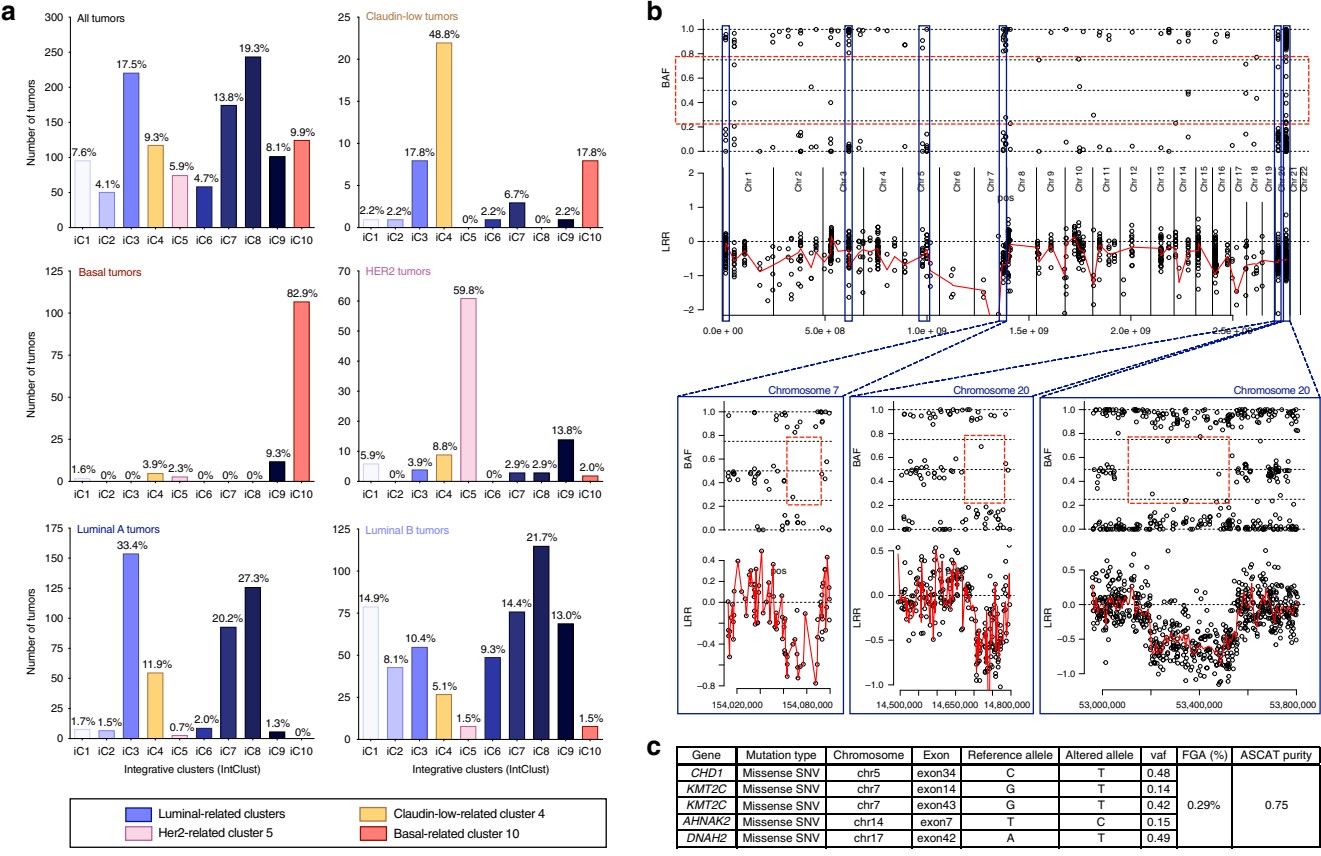

**Fig. 1 Claudin-low tumors are distributed across luminal-like IntClust3, basal-like IntClust10, and CNA-devoid IntClust4, which harbors rare focal genomic alterations. a** Repartition of breast tumors from the METABRIC cohort for each molecular subtype among integrative clusters. Number of tumors are represented on the y axis and the corresponding percentages are indicated above each bar. All tumors (n = 1266); claudin-low tumors (n = 45); basal tumors (n = 129); HER2 tumors (n = 102); luminal A tumors (n = 461); luminal B tumors (n = 429). **b, c** Identification of focal genomic alterations in a CNA-devoid IntClust4 claudin-low sample (MB-6052/FGA < 0.3%). **b** BAF and LRR plots of SNPs localized in genomic regions of copy number deletion (LRR < 0.4). **c** Mutation analysis from targeted sequencing data. BAF: B allele frequency; CNA: copy number alteration; FGA: fraction of genome altered; IntClust: integrative cluster; LRR: log R ratio; SNP: single-nucleotide polymorphism.

analysis of somatic mutations highlighted the presence of several point mutations with a variant allele frequency higher than 0.4 (Fig. 1c and Supplementary Fig. 3b). These two results unequivocally demonstrate the existence of claudin-low tumors that develop in the absence of gross CIN.

**Genomic diversity of claudin-low tumors**. Although most claudin-low tumors are diploid (Fig. 2a), the extent of FGA was highly variable across tumors (Fig. 2b), confirming their intrinsic diversity[24,25]. The use of Gaussian finite mixture models revealed a trimodal distribution, reflecting the existence of three subgroups of claudin-low tumors relative to their level of FGA (Fig. 2c, d). The subgroup 1 of claudin-low tumors (CL1), defined by a low FGA (<10%), was mostly composed of ER-negative tumors that were all stratified into integrative cluster 4 (Fig. 2d–f). The subgroup 2 (CL2) showed an intermediate level of FGA (>10% and <30%), similar to that of luminal A breast tumors. Consistent with this finding, CL2 was mostly composed of ER-positive tumors mainly stratified in two luminal-related clusters, whereas 35% of CL2 tumors belonged to cluster 4. The subgroup 3 (CL3) displayed a high FGA (>30%), similar to the one found in basal-like breast cancers. Consistent with this notion, 33% of CL3 tumors were classified into the genomically unstable cluster 10. However, this subgroup was heterogeneous, with both ER-negative or ER-positive tumors, and a significant fraction of CL3 tumors were found in luminal-related clusters and in cluster 4. Of

note, although these analyses were performed on tumors selected for their high tumor purity, similar data were obtained when studying the whole cohort of claudin-low tumors (Supplementary Fig. 4), further strengthening the characterization of the three claudin-low subgroups.

**Distinct gene expression signatures of claudin-low subgroups**. Gene-set enrichment analysis (GSEA) was next used to identify pathways differentially regulated among the three claudin-low subgroups, by comparing one with the other two (CL1 vs. CL2/3, CL2 vs. CL1/3, and CL3 vs. CL1/2). More than 15,000 pathways, available in H (Hallmarks), C2 (encompassing curated gene sets from literature and canonical pathways such as KEGG, REACTOME, or BIOCARTA), and C5 (comprising GO pathways) gene sets from Molecular Signature Database (MSigDB), were tested. These analyses highlighted as follows: (i) an enrichment in stem cell-related signatures in CL1, (ii) an enrichment in luminal-related signatures in CL2, and (iii) an enrichment in cell cycle-related pathways in CL3 tumors (Fig. 3a). Of note, global gene expression signatures demonstrated that the CL2 and CL3 subgroups showed lower luminal- and basal-related signature scores, respectively, compared with luminal- and basal-like breast tumors, while exhibiting a significant enrichment in invasiveness-related signatures (Supplementary Fig. 5a, b). We then performed single-sample GSEA (ssGSEA) analyses to determine the differentiation profile of the three claudin-low

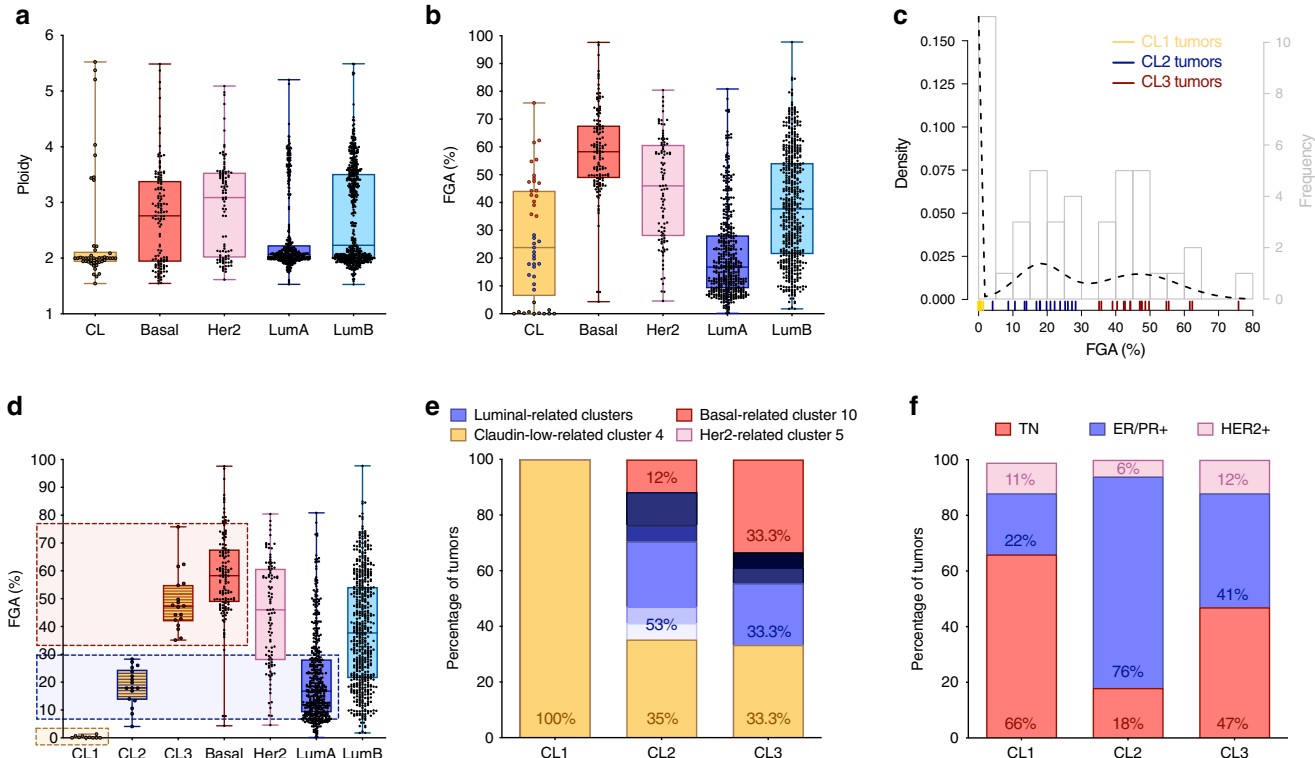

**Fig. 2 Claudin-low tumors form a genomically and molecularly heterogeneous subgroup. a** Ploidy for each molecular subtype of the METABRIC cohort. Claudin-low tumors are mainly diploid. **b** FGA% for each molecular subtype. Claudin-low tumors display overall low levels of genomic alterations. **c** Gaussian mixture model application on FGA distribution across claudin-low tumors reveals three CNA-related subgroups of tumors. Each bar on the $x$ axis corresponds to one claudin-low tumor. **d** FGA% in claudin-low subgroups compared with other molecular subtypes. **e** Integrative clusters and **f** breast cancer receptor status distribution in each claudin-low subgroup. The CNA-devoid CL1 subgroup is associated with IntClust4 and TN status, whereas CL2 and CL3 are respectively related to luminal (luminal clusters and hormone receptor expression) and basal-like (IntClust10 and TN tumor enrichment) subtypes. Respective colors of luminal-related clusters are shown in Fig. 1. Boxplot: center line, median; box limits, upper and lower quartiles; whiskers, minimum to maximum; all data points are shown. Claudin-low tumors ($n = 45$); basal tumors ($n = 129$); HER2 tumors ($n = 102$); luminal A tumors ($n = 461$); luminal B tumors ($n = 429$); CL1 tumors ($n = 10$); CL2 tumors ($n = 17$); CL3 tumors ($n = 18$). CL: claudin-low; TN: triple negative.

subgroups. Gene expression signatures for MaSCs, LPs, and mL cells were generated based on the transcription profiles of subpopulations of normal human mammary epithelial cells isolated from reduction mammoplasty tissues, as shown in Morel et al.[18] (Supplementary Data 1). Consistent with previous GSEA analysis, CL1 tumors showed a strong enrichment in the MaSC signature, whereas CL2 tumors exhibited a mL signature (Fig. 3b). Again, CL3 tumors showed a significant heterogeneity with a trend toward the LP signature. Notably, very similar data were obtained when using an alternative set of MaSC, LP, and mL signatures[14] (Fig. 3c).

**Identification of a gene expression-based classifier**. We next generated a gene expression-based classifier by using the nearest shrunken centroid method[26], with the objective of discriminating claudin-low subgroups. A claudin-low gene list of 137 genes discriminating the 3 claudin-low subgroups was thus created (Supplementary Fig. 6a, b). This expression-based classifier showed an accuracy of 91% when compared with the FGA-based classification using Gaussian finite mixture models (Supplementary Fig. 5c, d). The classifier was then applied to claudin-low samples from The Cancer Genome Atlas Network (TCGA; https://www.cancer.gov/about-nci/organization/ccg/research/structural-genomics/tcga) and the Cancer Cell Line Encyclopedia (CCLE[27]), and the level of FGA was used as a readout to verify the characteristics of each claudin-low subgroup. Validating the accuracy of our classifier, CL1, CL2, and CL3 tumors and cell

lines showed low, moderate, and elevated FGA levels, respectively (Supplementary Fig. 6e, f). Of note, we performed a pathway analysis of CL1 discriminant genes that revealed an enrichment in stem cell and pediatric cancer markers (Supplementary Fig. 6g).

**Distinct gene methylation profiles of claudin-low subgroups.** Differentially methylated gene analysis was next performed in claudin-low subgroups followed by pathway-enrichment analysis, using over 15,000 pathways available in MSigDB (HALLMARKS, C2, and C5 gene sets) (Supplementary Fig. 7). When compared with CL2 and CL3, the CL1 subgroup presented 40 differentially methylated genes, and among them 75% were hypomethylated and enriched in stemness and EMT markers (Fig. 4a). The CL2 subgroup encompassed 68 differentially methylated genes compared with CL1 and CL3, and among them 88% were hypermethylated and enriched in basal-like and EMT pathways (Fig. 4b). Finally, the CL3 subgroup was characterized by 219 differentially methylated genes compared with CL1 and CL2; among them, 58% were hypermethylated and related to luminal pathways and 42% presented a hypomethylation profile and were associated with basal-like pathways (Fig. 4c). To correlate these distinct gene methylation profiles with gene expression, we performed a gene expression analysis of stemness and EMT gene markers in all three claudin-low subgroups and in the intrinsic subtypes of breast cancers from METABRIC, TCGA, and CCLE cohorts. The CL1 subgroup had the highest score for stemness- (*ALDH1A1*, *PROCR*) and EMT- (mesenchymal markers: *ZEB1*,

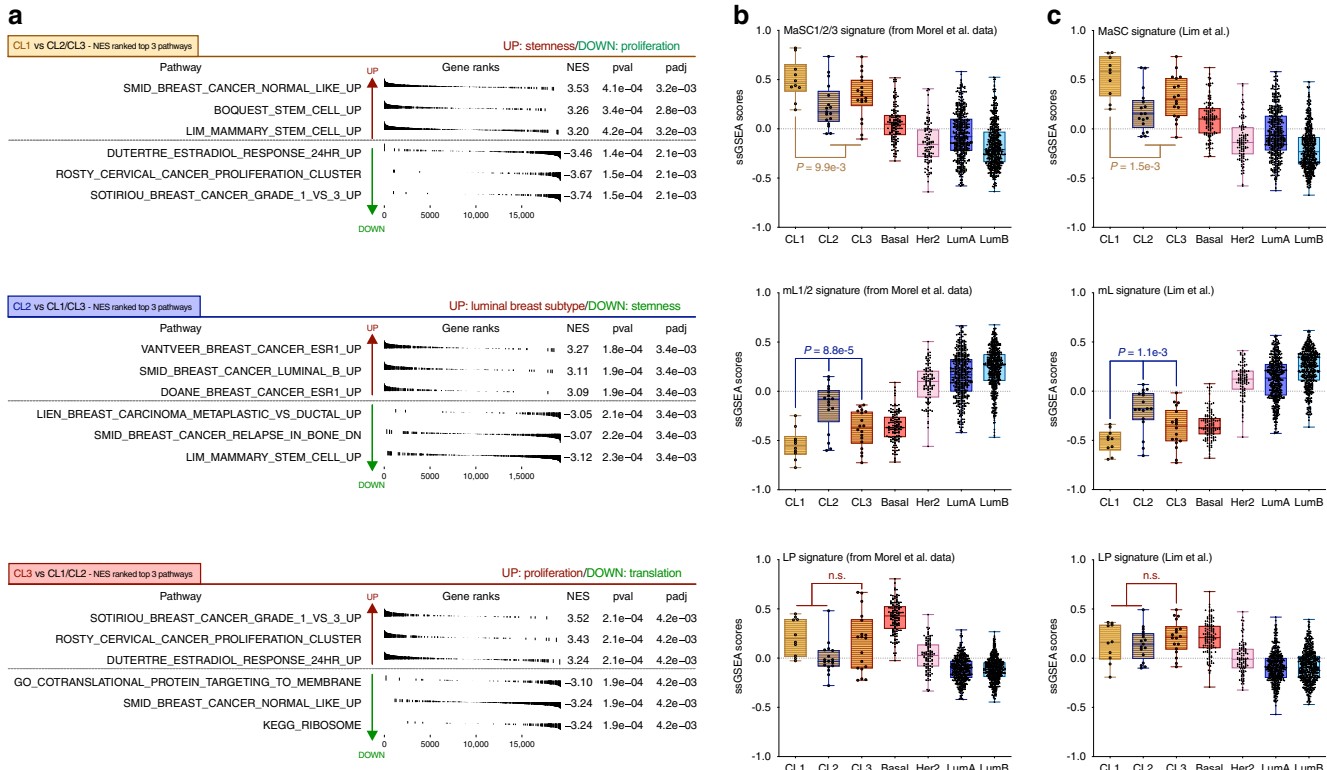

**Fig. 3 Claudin-low subgroups show distinct gene expression signatures relative to their differentiation status. a** GSEA comparing global gene expression of each claudin-low subgroup from the METABRIC cohort to the two others. The three pathways with the highest and lowest enrichment scores are represented (>15,000 tested pathways—NES ranking). **b, c** Gene expression analysis for each molecular subtype of (**b**) MaSC1/2/3, LP, and mL1/2 transcriptomic signatures (generated from Morel et al.[18] data) and (**c**) previously published MaSC, LP, and mL transcriptomic signatures (from Lim et al.[14] publication). The CL1 subgroup displays strong stemness features, whereas CL2 and CL3 present luminal- and basal-like transcriptomic characteristics, respectively. Wilcoxon tests. Boxplot: center line, median; box limits, upper and lower quartiles; whiskers, minimum to maximum; all data points are shown. Basal tumors ($n = 129$); HER2 tumors ($n = 102$); luminal A tumors ($n = 461$); luminal B tumors ($n = 429$); CL1 tumors ($n = 10$); CL2 tumors ($n = 17$); CL3 tumors ($n = 18$). GSEA: gene-set enrichment analysis; LP: luminal progenitor; MaSC: mammary stem cell; mL: mature luminal; NES: normalized enrichment score.

*VIM*, *CDH2*; epithelial markers: *EPCAM*, *CDH1*) related genes, whereas CL2 and CL3 displayed an intermediate score compared with CL1 tumors and with their relative luminal/basal counterparts (Supplementary Fig. 8).

**Distinct oncogenic pathways in claudin-low subgroups**. To characterize the biological pathways driving the development of the three subgroups of claudin-low tumors, we determined, for each breast tumor sample from METABRIC and TCGA cohorts, ssGSEA scores for all MSigDB hallmark gene-set signatures. We then conducted an unsupervised clustering based on the median score of each of the 48 biological processes computed per breast molecular subtype (Fig. 5). Interestingly, in both datasets, the three claudin-low subgroups clustered together in a specific branch of the dendogram, which further discriminated CL1 from the CL2/CL3 tumors, highlighting the specific biological processes distinguishing claudin-low from non-claudin-low and CL1 from the CL2/CL3 tumors. Furthermore, unsupervised pathway clustering identified specific biological functions differentially enriched in claudin-low, basal, and luminal breast tumors. Among the pathways specifically enriched in all claudin-low subgroups, some of them were linked to immunity function, supporting the hypothesis of privileged infiltration by immune cells within the microenvironment of claudin-low tumors[4,19]. Additional gene expression analyses performed on METABRIC and TCGA

cohorts using two different cellular signature deconvolution algorithms confirmed that these tumors were highly infiltrated by immune cells (Supplementary Fig. 9a–d). Nevertheless, unsupervised clustering of microenvironment signatures (immune and non-immune cell subsets) did not allow to discriminate the different claudin-low subgroups (Supplementary Fig. 9e–h). The additional pathways specifically enriched in all claudin-low subgroups included the TP53-dependent pathway, the mitogen-activated protein kinase (RASMAPK) signaling pathway and a differentiation/EMT-related pathway (Fig. 5). CL1 presented the highest enrichment in claudin-low-related pathways, when compared with all other subtypes. Of note, the CL2 subgroup displayed a concomitant enrichment in claudin-low and luminal pathways, whereas the CL3 subgroup was enriched in claudin-low and basal pathways (Fig. 5).

To validate these observations, we generated ssGSEA scores for the pathways previously identified in Fig. 5 (EMT, RAS/MAPK, proliferation, estrogen, and DNA damage response (DDR)), using MSigDB C2 curated gene sets (KEGG, GO, REACTOME…). The same pathways were also analyzed at the protein level using reverse-phase protein array (RPPA) data from TCGA[28]. Supporting our previous results, CL2 and CL3 showed, among the three claudin-low subgroups, the highest score for luminal- (estrogen-related response) and basal-related (proliferation and DDR) signatures, respectively (Supplementary Fig. 10). Nevertheless, statistical analyses revealed a significantly attenuated phenotype

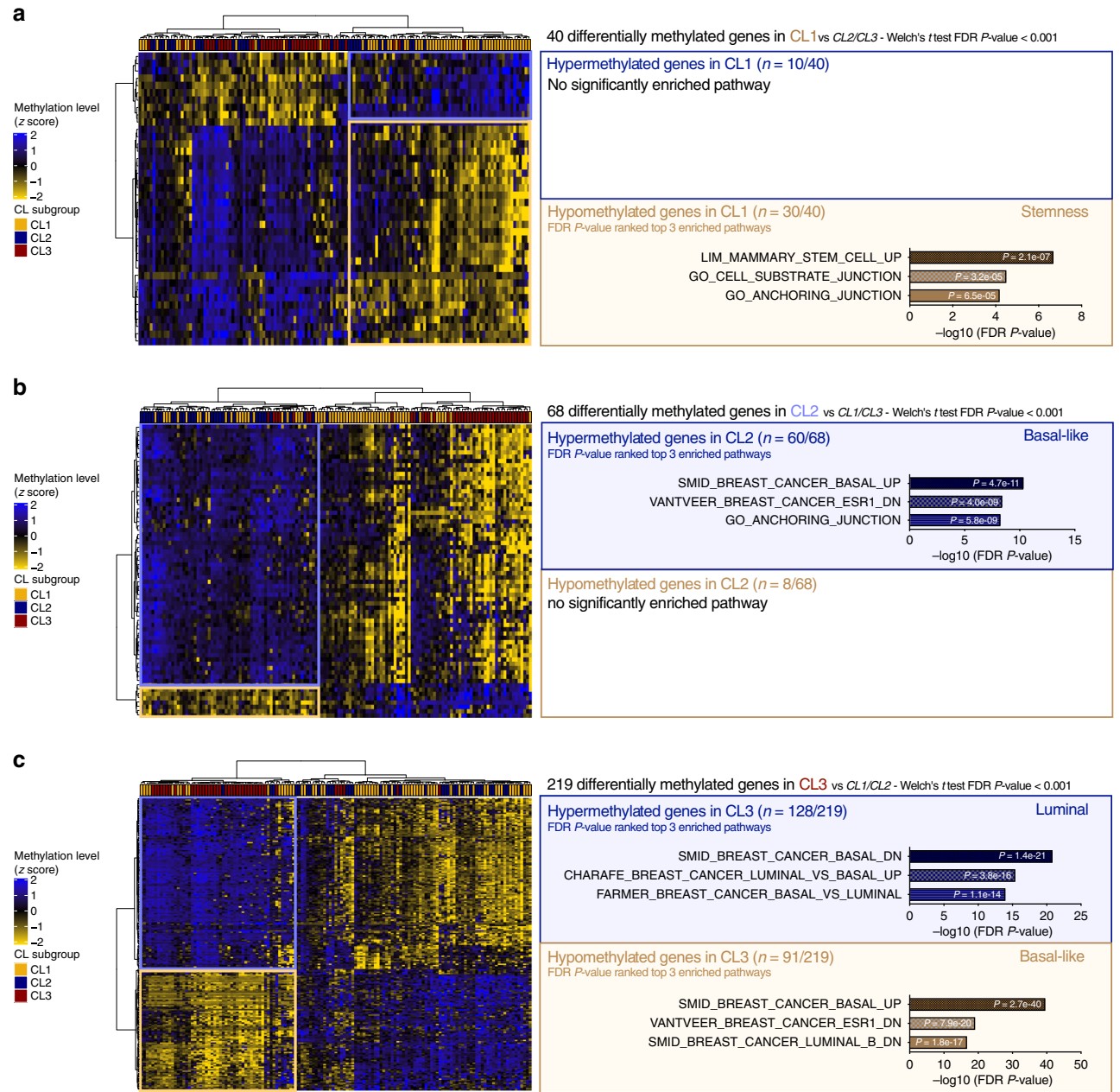

**Fig. 4 Claudin-low subgroups show distinct gene methylation profiles relative to their differentiation status.** Heatmaps and pathway-enrichment analysis of differentially methylated genes between **a** CL1 vs. CL2 and CL3 tumors from TCGA, **b** CL2 vs. CL1 and CL3 tumors from TCGA, and **c** CL3 vs. CL1 and CL2 tumors from TCGA. CL1 tumors mainly display hypomethylated genes that are related to a stemness phenotype, CL2 tumors are overall hypermethylated for genes associated with basal-like features, and CL3 tumors show more heterogeneous methylation profiles with both hypermethylated genes (linked to luminal phenotype) and hypomethylated genes (relative to basal-like characteristics) (see Supplementary Fig. 5 for methodology). Clustering method: Ward's; distance: Spearman. FDR: false discovery rate.

compared with their related molecular subtype. Furthermore, both CL2 and CL3 showed an activation of the EMT process (Fig. 6a, b, g), coherent with the downregulation of hormone (in CL2) and proliferation/DDR (in CL3) signatures, compared with luminal and basal tumors, respectively (Supplementary Fig. 10 and Fig. 6g). Moreover, CL1 displayed a low proliferation activity and a low DDR (Supplementary Fig. 10a, b, e, f), as well as a strong TP53 signaling pathway signature associated with a low frequency of *TP53* mutations (Fig. 6c, d, g). Finally, CL1 exhibited a high activation of the RAS-MAPK signaling pathway, a feature also found in CL2 and CL3, although at a lower level (Fig. 6e–g).

Consistent with these findings, claudin-low cell lines were significantly more sensitive to MEK inhibitors (MEKi) than non-claudin-low cell lines (Fig. 6h).

Altogether, our data strongly support the notion that the CL1 subgroup of claudin-low tumors was related to normal MaSCs, CL2 to normal mL cells and to luminal breast cancers, and CL3 to basal-like breast cancers. Unstratified claudin-low tumors have been previously associated with poor prognosis[4,29]. When stratified by CL subgroups, variations in survival outcome were found, consistent with their presumed origin (Supplementary Fig. 11). Indeed, the basal-like-related CL3 subgroup of

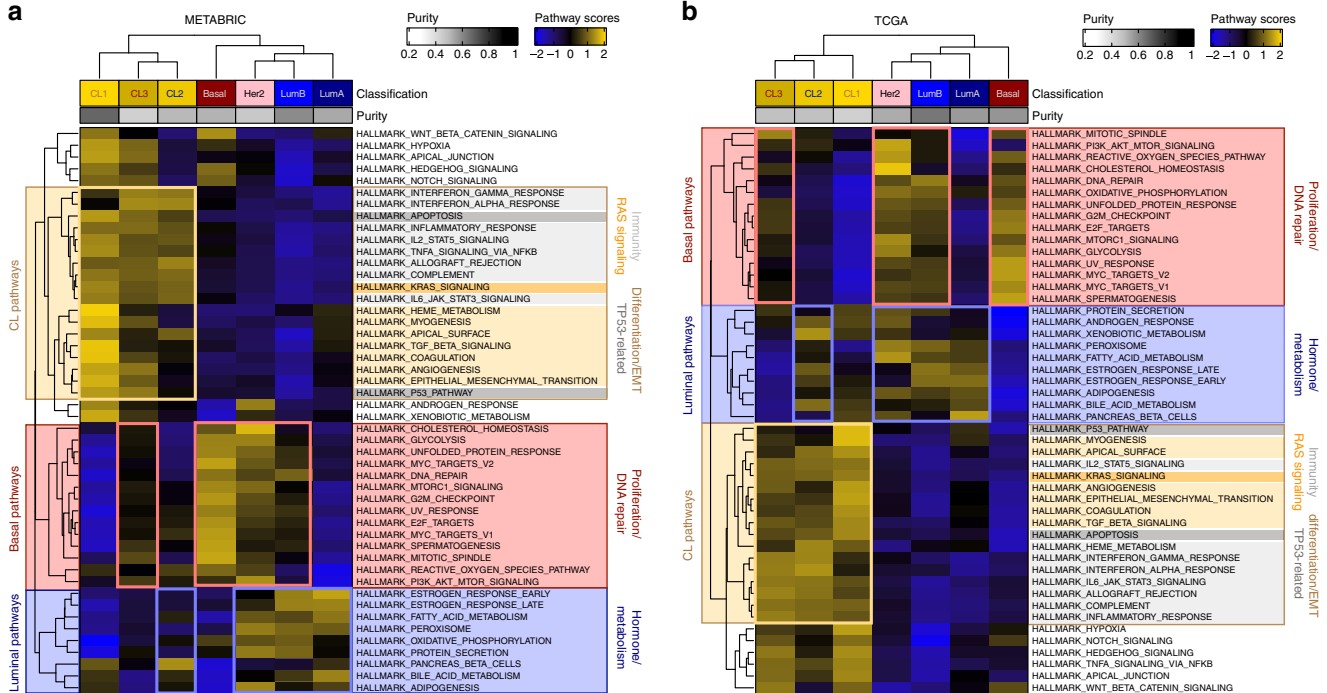

**Fig. 5 Claudin-low subgroups show distinct activated biological pathways.** Heatmaps of ssGSEA scores for each MSigDB hallmark gene-set signature (median by molecular subtype) for **a** METABRIC and **b** TCGA datasets. ASCAT median purity score of each subtype is indicated on the top heat map annotation. Claudin-low tumors cluster together in a specific branch of the dendrogram downstream distinguishing CL1 from CL2/CL3 tumors. Hallmark gene-set cluster in three main groups of pathways related to luminal, basal, and claudin-low tumors. ASCAT: allele-specific copy number analysis of tumors; MSigDB: molecular signature database; ssGSEA: single-sample GSEA. Clustering method: Complete; distance: Euclidean.

claudin-low tumors was associated with low overall and disease-free survivals, as compared with the luminal-related CL2 subgroup. In line with their low proliferation index and their low frequency of *TP53* mutations, tumors of the CL1 subgroup were associated with a favorable prognosis.

## Discussion

Here we have characterized the intrinsic diversity within the claudin-low breast cancers by demonstrating the existence of three main molecular subgroups. Consistent with the data from a recent study[30], published during the revision process of our manuscript, we have shown that these subgroups are associated with distinct survival outcomes. Moreover, our study has delved into the underlying reasons of the observed heterogeneity, revealing that claudin-low tumors exhibit different developmental origins.

Supporting the prevailing hypothesis that some of these tumors are generated from basal-like breast cancers, the comprehensive characterization of METABRIC and TCGA databases have revealed the existence of claudin-low breast cancers (CL3 subgroup) with characteristics of basal mammary epithelial cells and of basal-like breast cancers. These traits include the hypomethylation of genes related to basal-related pathways, the hypermethylation of genes related to luminal-related pathways, a high level of expression of proliferation-related genes and a strongly disturbed genomic landscape. Several experimental data substantiate the hypothesis of the basal origin of claudin-low breast cancers, including the demonstration that the oncogenic transformation of basal mammary epithelial cells and of LPs can generate malignant cells with mesenchymal characteristics through the activation of an EMT process[17,31]. However, highlighting a first degree of heterogeneity, a significant fraction of claudin-low tumors (CL2 subgroup) displays a gene expression signature of mL cells and shares common genetic, epigenetic and

transcriptomic features with the luminal A intrinsic subtype. Although unexpected, this finding is reminiscent of the experimental observation that EMT induction in luminal breast cancer cell lines confers them with a claudin-low expression pattern[32]. A second degree of heterogeneity is shown by the CL1 subgroup of claudin-low breast cancers. Among the three subgroups of claudin-low tumors, CL1 appears as the most distinct, with differential transcriptomic, epigenetic, and genetic traits that comprise a prominent stemness signature and a paucity of genomic aberrations. Although TNBCs with negligible CNAs were previously reported[21,22], their existence was questioned due to the marked immune and stromal cell infiltration attributed to claudin-low tumors[11]. Here, the identification of focal genetic alterations in tumors selected for their purity unequivocally demonstrates the existence of CNA-devoid claudin-low tumors. Beyond their stemness characteristics and their low CIN, CL1 tumors have additional characteristic features when compared to all other breast cancers. These traits include (i) a high expression of the ZEB1 EMT-inducing transcription factor and of its target, the methionine sulfoxide reductase MSRB3; (ii) a frequent activation of the RAS-MAPK signaling pathway; (iii) a low activation of the DDR; and (iv) a low frequency of *TP53* mutations (Supplementary Figs. 8 and 10, and Fig. 6). These findings are highly consistent with the stemness features of CL1 tumors. Indeed, we demonstrated recently that normal human MaSCs exhibit a preemptive antioxidant program driven by ZEB1 and MSRB3, providing them with the unique capacity to readily adapt to an aberrant activation of the RAS-MAPK pathway[18]. As a consequence, RAS-driven malignant transformation of MaSCs occurs in the absence of DNA damage, thereby alleviating the selective pressure on the inactivation of the TP53-dependent failsafe program[8,18]. Altogether, these findings led us to propose a model with two alternative paths leading to the development of claudin-low tumors (Fig. 7). A direct path relies upon the malignant

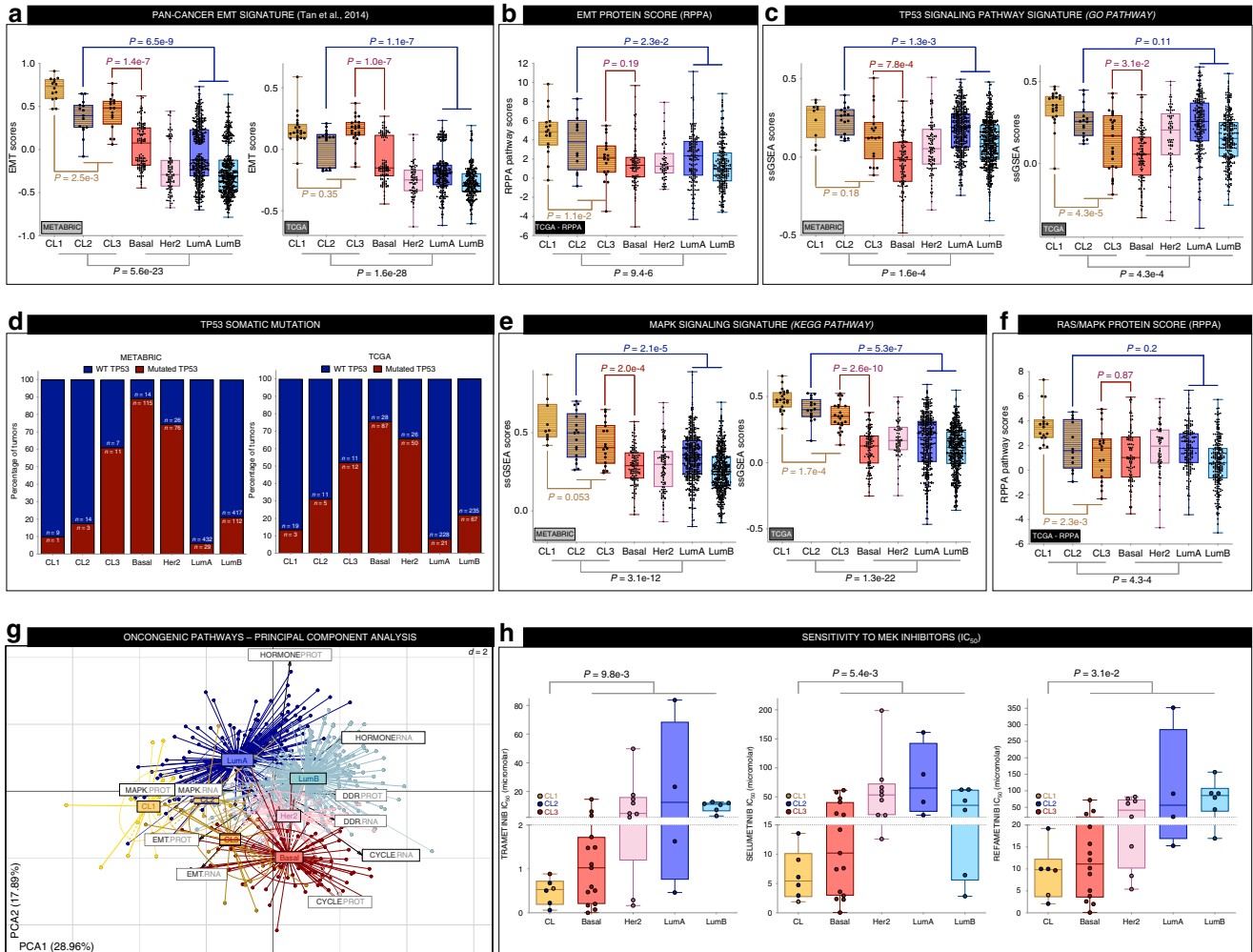

**Fig. 6 Claudin-low subgroups jointly display high EMT features and MAPK pathway activation. a**, **c**, **e** ssGSEA score distribution for each molecular subtype, from METABRIC and TCGA cohorts, related to **a** EMT, **c** TP53, and **e** RAS/MAPK pathways. **d** *TP53* somatic mutation distribution across breast molecular subtypes of METABRIC and TCGA cohorts. **b**, **f** Protein-based pathway scores (RPPA data) for each molecular subtype from TCGA breast tumors related to **b** EMT and **f** RAS/MAPK. **g** Principal component analysis (PCA) of TCGA breast tumors according to EMT, RAS/MAPK, cell cycle/proliferation, hormone-related and DDR pathways at RNA (ssGSEA scores), and protein (RPPA scores) levels. For each sample, molecular subtype is indicated in color and variables (pathways) are highlighted in black squares. **h** IC50 distribution according to molecular subtype for three different MEK inhibitors available in the GDSC database. Claudin-low cell lines commonly exhibit higher sensitivity to MAPK inhibitors than other breast molecular subgroups. Wilcoxon tests. Boxplot: center line, median; box limits, upper and lower quartiles; whiskers, minimum to maximum; all data points are shown. TP53 signaling pathway signature: GO positive regulation of signal transduction by p53 class mediator; MAPK signaling signature: KEGG MAPK signaling pathway. DDR: DNA damage response; EMT: epithelial-mesenchymal transition; RPPA: reverse-phase protein array.

transformation of a normal MaSC, leading to undifferentiated tumors with a low genomic instability and a low frequency of *TP53* mutations. An indirect path relies upon the activation of an EMT process, over the course of tumor progression. Beyond a certain extent of transdifferentiation, the gain of mesenchymal features triggers the conversion into a gene expression signature of claudin-low tumor subtype. When it occurs in a basal-like tumor, EMT promotes the evolution toward a claudin-low tumor with extensive genomic aberrations as the reflect of previous periods of exacerbated CIN. When this process occurs in a luminal breast cancer, it leads gradually to a claudin-low tumor with a moderate level of genomic aberrations.

Overall, this model illustrates the emerging concept of cellular pliancy[8]. The level of pliancy is defined by the intrinsic suscepti-bility of a discrete cell state to undergo malignant transformation or degeneration after sustaining a particular oncogenic insult[8,33]. In this regard, the low genomic instability of CL1 tumors might

thus reflect the high pliancy of human MaSCs for RAS activation. Interestingly, although CL1 tumors show the most frequent activation of the RAS-MAPK pathway, it is noteworthy that the activation of this mitogenic pathway is also a common event in CL2 and CL3 tumors, compared with other breast cancer sub-types. This latter finding may reflect the capacity of the RAS signaling pathway to induce EMT in permissive conditions[34–36]. Implicit in this notion is that the activation of the RAS-MAPK pathway may be a primary oncogenic event in the tumorigenic process leading to CL1 tumors and a secondary event in the course of tumor progression in a fraction of basal-like and luminal breast cancers, eventually leading to CL2 and CL3 tumors.

Although this hypothesis remains to be tested, our data may have significant clinical implications in light of the previously described resistance of EMT-related tumors to a variety of anti-cancer therapies[37–39]. Indeed, the identification of the recurrent

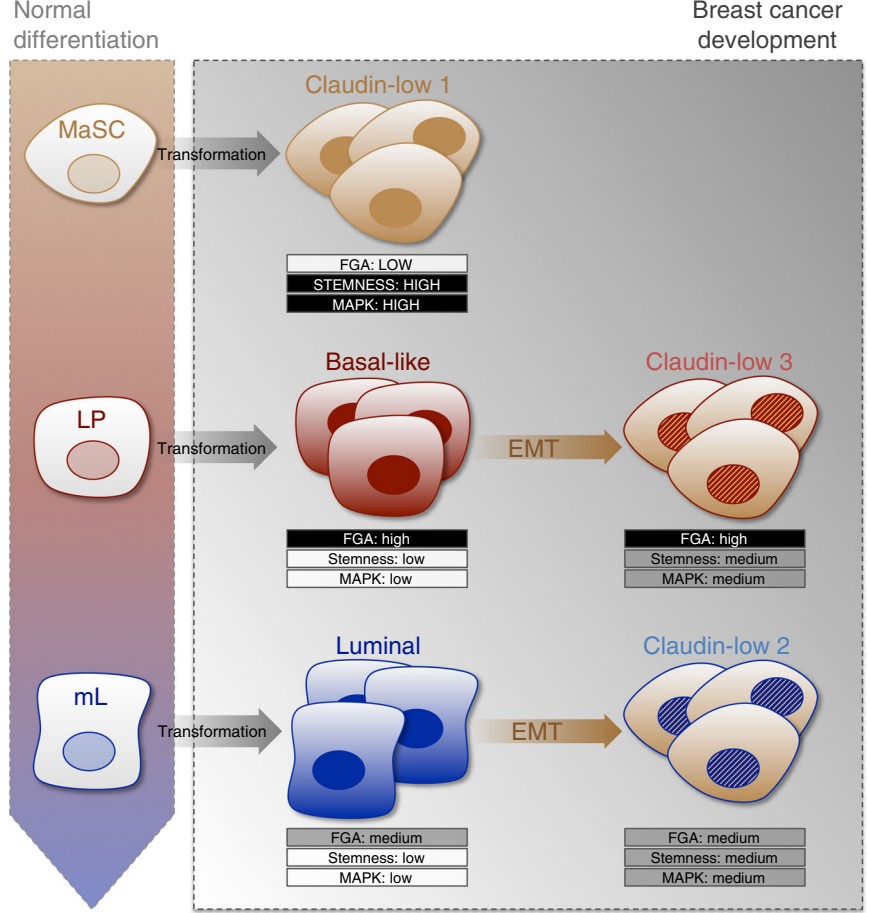

**Fig. 7 The diversity of claudin-low tumors reflects the impact of the cell-of-origin on the evolutionary process of breast tumorigenesis.** The malignant transformation of a normal MaSC leads to undifferentiated tumors with a low genomic instability and a low frequency of *TP53* mutations. These features are characteristic of the CL1 subgroup. The activation of an EMT transdifferentiation process and the gain of mesenchymal features in a basal-like tumor promotes the development of a CL3 tumor with extensive genomic aberrations. EMT commitment in a luminal-like breast cancer leads gradually to a CL2 tumor with a moderate level of genomic aberrations.

activation of the RAS-MAPK pathway in claudin-low breast cancers may lead to new therapeutic opportunities that need to be tested in preclinical models.

## Methods

**Samples**. Breast tumor samples used in this study were from METABRIC[21] and TCGA Research Network (https://www.cancer.gov/tcga) cohorts, and breast cancer cell lines were from CCLE database[27].

**Statistics**. All analyses and statistical tests were carried out with the R software (version 3.6.1)[40]. Heatmaps were generated with ComplexeHeatmap, principal component analysis was completed with ade4, Gaussian finite models were performed with mclust and Rmixmod, and survival analyses were conducted with survival R packages[25,41–43]. All statistical tests were two-tailed. Figures were created using either the R software or GraphPad Prism 8.0 (GraphPad Software, Inc., San Diego, USA).

**Expression data processing**. METABRIC microarray expression data from discovery and validation sets were extracted from the EMBL–EBI archive (EGA, http://www.ebi.ac.uk/ega/; accession number: EGAS00000000083) (Normalized expression data files)[21]. Normalized expression data per probe of the discovery set and the validation set were combined after normalization of each set independently with a median $Z$-score calculation for each probe. The expression levels of different probes associated with the same Entrez Gene ID were averaged for each sample in order to obtain a single expression value by gene.

The Cancer Genome Atlas Breast Invasive Carcinoma (TCGA BRCA) RNA sequencing (RNASeq) expression data were extracted as FPKM (fragments per kilobase of transcript per million mapped reads) values from the Genomic Data Commons (GDC) data portal (https://portal.gdc.cancer.gov/). FPKM data by gene were converted to transcripts per kilobase million (TPM) as follows: for each gene

$g \in G$ and each sample $s \in S$,

$$\text{TPM}(g, s) = \left( \frac{\text{FPKM}(g, s)}{\sum_{i=1}^{G} \text{FPKM}(i, s)} \right) \times 10^6$$

RNASeq expression data from the CCLE breast cell lines were extracted as RPKM values from the CCLE data portal (https://portals.broadinstitute.org/ccle). RPKM data by gene were converted to TPM as follows: for each gene $g \in G$ and each sample $s \in S$,

$$\text{TPM}(g, s) = \left( \frac{\text{RPKM}(g, s)}{\sum_{i=1}^{G} \text{RPKM}(i, s)} \right) \times 10^6$$

**Molecular breast cancer subtype assignment**. Triple-negative (TN) status was determined from clinical data obtained through the Synapse platform (https://www.synapse.org/) (syn1757053) for METABRIC dataset and the GDC data portal (https://portal.gdc.cancer.gov/) for TCGA dataset. Expression value for estrogen and progesterone receptors were combined with the SNP6 state for *ERBB2* gene, to define the TN status of each available tumor.

Integrative cluster and breast cancer molecular subtype attribution were performed using the R package genefu[44]. Basal-like, luminal A, luminal B, Her2, and normal-like subtype assignments were computed from five different algorithms (PAM50, AIMS, SCMGENE, SSP2006, and SCMOD2)[45–49]. An assignment was considered final if defined by at least three different algorithms. In case of divergence between classifiers, PAM50 subtype attribution was used.

The claudin-low subtype classification was defined by the nearest centroid method. For that, the Euclidean distance between each tumor sample (from

METABRIC and TCGA cohorts) and the previously described claudin-low and non-claudin-low centroids for tumor samples were determined, using the 1667 genes defined by Prat et al.[4] as significantly differentially expressed between claudin-low tumors and all other molecular subtypes. The tumor-related centroids (differentially expressed genes between claudin-low and non-claudin-low tumors using significance analysis of microarrays) were used rather than the cell-line-related centroids (nine cell lines), as cell culture dramatically changes tumor phenotype and thus transcriptomic signature of primary samples. The use of a high-purity threshold for tumor selection allowed to circumvent the contamination bias related to the tumor-related centroid (Supplementary Fig. 1). For CCLE breast cancer cell lines, claudin-low status assignment was performed using the nine-cell-line predictor via the R package genefu[44].

**Pathway enrichment analysis**. All pathway-enrichment analyses were conducted using MSigDB gene sets H, C2, and C5 from msigdbr R package[50]. GSEAs were carried out using fgsea R package[51]. Gene lists were pre-ranked using Signal2Noise metric. ssGSEA scores were computed through gsva R package[52]. Pan-cancer transcriptomic EMT signature, defined by Tan et al.[53], was used to compute EMT score for each sample.

**Normal mammary cell transcriptomic signatures**. Previously published micro-array expression data were used to generate lists of differentially expressed genes along the mammary epithelial cell hierarchy (MaSC1/2/3, LP, and mL1/2)[18]. Microarray data were robust multiarray average-normalized through oligo R package and differential expression analysis was performed using limma R package[54,55]. The top 500 false discovery rate (FDR) p-value ranked genes differentially expressed between one subpopulation (MaSC or LP, or mL cells) vs. the two others (FDR p-value < 0.05) were used as transcriptomic signature for each of the three cell subpopulations (Supplementary Data 1).

**Claudin-low subgroups classifier**. Expression-based classifier for the three claudin-low subgroups was identified using shrunken nearest centroid method through pamr R package[26]. A 1.87 threshold for centroid shrinkage was defined after examination of training errors and the cross-validation results. Finally, the nearest shrunken centroid classifier encompassed 137 genes whose expression discriminate the three FGA-related claudin-low subgroups in METABRIC cohort (Supplementary Fig. 6).

**Microenvironment analysis**. Estimation of immune and non-immune cell fractions from tumor microenvironment were determined through gene expression analysis using Immunedeconv R package[56] together with Xcell and MCPCounter deconvolution methods[57,58].

**Copy number data processing**. METABRIC segmented copy number data from discovery and validation sets were extracted from the EGA (http://www.ebi.ac.uk/ega/; accession number: EGAS00000000083) (Segmented (CBS) copy number aberrations (CNA) files)[21]. TCGA BRCA segmented copy number data were extracted from the GDC data portal repository (files corresponding to alignments on the hg19 version of the human genome without germline CNV were chosen). FGAs was evaluated from TCGA and METABRIC segmented copy number data (both generated from Affymetrix SNP6.0 arrays) as follows:

$$\text{FGA} = \frac{\sum_{\text{CN}i > \text{WM}+T} L(i)}{\left(\sum L(i)\right)} + \frac{\sum_{\text{CN}i < \text{WM}-T} L(i)}{\left(\sum L(i)\right)}$$

For each segment $i$, $\text{CN}_i$ is the mean LRR along segment $i$, $L(i)$ is the length of segment $i$, WM is the weighted median of $\text{CN}_i$ by $L(i)$ for each sample $I$, and $T$ is the threshold value of the $\text{CN}_i$ above which the segments are considered to be altered. In other words, FGA is the ratio of the sum of the lengths of all segments with signal above the threshold to the sum of all segment lengths. For CCLE cell lines analysis, $T$ was set as 0.2, whereas for METABRIC and TCGA tumors analysis, $T$ was set as 0.1.

ASCAT ploidy and purity estimates were extracted from COSMIC data repository (https://cancer.sanger.ac.uk/cosmic/)[20]. To avoid any substantial bias due to non-tumor cell contamination, tumors without available estimation of ASCAT aberrant tumor cell fraction were removed from the whole cohort. Furthermore, a stringent purity threshold (ASCAT purity > 0.38) was determined, by applying Wilcoxon test, to select purest tumor samples from TCGA and METABRIC databases when applicable.

**Methylation data processing**. Normalized methylation β-values per gene were extracted from cBioPortal (http://www.cbioportal.org/)[59,60]. As previously described for genome-wide expression–methylation quantitative trait loci analysis[61], genes for which methylation β-values were negatively correlated with expression values (Spearman's ρ < 0 and FDR p-value < 0.05) were selected for differential methylation analysis (Supplementary Fig. 7).

**RPPA data processing**. RPPA level 4 data were extracted from the cancer proteome atlas portal (https://tcpaportal.org/tcpa/download.html)[62]. RPPA pathway lists, defined by Akbani et al.[28], were used to compute protein pathway scores for each sample.

**Somatic mutation data**. METABRIC somatic mutation data from targeted sequencing were obtained from Pereira et al.[63] (https://github.com/cclab-brca/mutationalProfiles/tree/master/Data) and TCGA somatic mutation data from whole-exome sequencing were obtained from Ellrott et al.[64]. Only somatic mutations annotated as exonic, in-frame, and non-silent were used for mutation analysis.

**Drug response data**. Predicted IC50 data for MEKi (Trametinib, Selumetinib, Refametinib) were download from Genomics of Drug Sensitivity in Cancer (GDSC) database (https://www.cancerrxgene.org/)[65].

**Clinicopathological analysis**. Complete clinical data were obtained through the Synapse platform (https://www.synapse.org/) (syn1757053) for METABRIC dataset.

For TCGA dataset, survival data were extracted from cBioPortal (TCGA BRCA, PanCancer Atlas) (http://www.cbioportal.org/) and other clinical data were obtained from the GDC data portal (https://portal.gdc.cancer.gov/)[59,60].

**Reporting summary**. Further information on research design is available in the Nature Research Reporting Summary linked to this article.

## Data availability

The complete set of CEL files from Morel et al.[18] is available in the GEO database under accession number GSE56031 and the ArrayExpress database under accession number E-MTAB-4145. METABRIC data are available in the EMBL–EBI archive (accession number: EGAS00000000083) and from Supplementary Information in Curtis et al.[21] and Pereira et al.[63]. TCGA BRCA RNAseq expression data were extracted as FPKM values from the GDC data portal (https://portal.gdc.cancer.gov/). RNAseq expression data from the CCLE breast cell lines were extracted as RPKM values from the CCLE data portal (https://portals.broadinstitute.org/ccle). Triple-negative (TN) status was determined from clinical data obtained through the Synapse platform (https://www.synapse.org/) (syn1757053) for METABRIC dataset and through the GDC data portal (https://portal.gdc.cancer.gov/) for TCGA dataset. METABRIC segmented copy number data from discovery and validation sets were extracted from the EGA (http://www.ebi.ac.uk/ega/; accession number: EGAS00000000083) (Segmented-CBS copy number aberration (CNA) files). TCGA BRCA segmented copy number data were extracted from the GDC data portal (https://portal.gdc.cancer.gov/). ASCAT ploidy and purity estimates were extracted from COSMIC data repository (https://cancer.sanger.ac.uk/ cosmic/). Normalized methylation β-values per gene were extracted from cBioPortal (http://www.cbioportal.org/). RPPA level 4 data were extracted from The Cancer Proteome Atlas (TCPA) portal (https://tcpaportal.org/tcpa/download.html). METABRIC somatic mutation data from targeted sequencing were obtained from Pereira et al.[63] (https://github.com/cclab-brca/mutationalProfiles/tree/master/Data) and TCGA somatic mutation data from whole-exome sequencing were obtained from Ellrott et al.[64]. Predicted IC50 data for MEKi (Trametinib, Selumetinib, Refametinib) were download from Genomics of Drug Sensitivity in Cancer (GDSC) database (https://www.cancerrxgene.org/). Complete clinical data were obtained through the Synapse platform (https://www.synapse.org/) (syn1757053) for METABRIC dataset. For TCGA dataset, survival data were extracted from cBioPortal (Breast Invasive Carcinoma, TCGA, PanCancer Atlas) (http://www.cbioportal.org/) and other clinical data were obtained from the GDC data portal (https://portal.gdc.cancer.gov/).

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

## Acknowledgements

We thank B. Manship for proofreading the manuscript. This work was supported by funding from the Ligue Nationale contre le Cancer (EL2016.LNCC/AIP and EL2019. LNCC/AIP) and SIRIC LYriCAN (INCa-DGOS-Inserm_12563). This work was additionally supported by funding from Inserm in the framework of the International Associated Laboratory between Cancer Research Centre of Lyon, France, and the Victorian Comprehensive Cancer Centre of Melbourne, Australia (Appel à projets LIA/LEA 2016, ASC17019CSA). This work was additionally supported by funding from *Le Cancer du sein. Parlons-en!* association.

## Author contributions

R.M.P. performed all bioinformatics and statistical analyses, prepared figures and methods, and assisted in data interpretation. M.O. and A.S. assisted in preparation of figures and manuscript writing. L.T., J.K., E.T., A.F., A.-S.S., F.H., and P.M. assisted in data interpretation. A.T. and A.-P.M. provided scientific content. F.H. provided critical evaluation of the manuscript and scientific content. A.P. conceived the project, designed analyses, interpreted data, and wrote the manuscript. All authors read and approved the final version of the manuscript.

## Competing interests

The authors declare no competing interests.
