## [Peer Review File · Nature Communications]

Reviewers' Comments:

Reviewer #1:

Remarks to the Author:

The work of Pommier et al., describes a multiomics approach to analyze a rare/elusive subtype of breast cancer: claudin-low and to infer the cell of origin of this molecular class.

In performing a comparison to non-Claudin-low tumors the authors, exclude the Claudin-low tumors with 'outlier' tumor purity very early in their analyses. They just choose to analyze the remaining 45 claudin-low tumors with sufficiently high tumor purity. Given that tumor subtypes are increasingly characterized by their microenvironment, this first step in the analysis is discussable.

- First, it means that the authors choose to analyze only 30% of the Claudin-low; should one not analyze all of them? Claudin-low represent approx. 10% of breast cancer cases, choosing to focus on only 30% of these 10% brings us to a very little proportion of breast cancer cases. While the authors further manage to subdivide the 45 cases further analyzed into three subgroups, one can question the relevance of such results when they are not verified in other cohorts than the METABRIC.

- Second it appears that in general Claudin-low have very low tumor-percent. Have the authors checked whether within the discovered here subgroups of Claudin-low do not differ tumor percent? by this I mean: do Claudin-low in IC3 have same tumor percent than non-claudin low in IC3, same for IC4 and IC10? While the authors use all non-claudin low tumors to exclude the Claudin low with the lowest tumor percent, the rest of the 30% Claudin-low analyzed (the one with decent purity) are not called claudin-low because of a lower tumor percent according to the main subtype they 'derive' from?

In other words: from the literature appears that the definition of claudin low subtype is intermingled with the microenvironment, so excluding it by setting filter on tumor purity the authors risk to produce skewed results.

Concerning the GSEA and the ssGSEA, would the authors have obtained the same results by comparing normal breast tissue, versus luminal versus basal tumors? Here, a more detail comparison with known subtype Luminal / Basal is warranted. The independence and originality of the CL subtypes is not clear in terms of pathway enrichment in Figure 3.

The authors chose the Gaussian mixture model of the FGA as the best way to find three distinct subgroups for Claudin-low. What is the justification for that? Other unbiased approaches could have been used such as unsupervised clustering of gene expression. Is the subdivision of the Claudin-low by PAM50 or integrative clusters a sufficient and relevant subdivision? Are the Claudin-low subtypes observed by the authors clinically-relevant or does it bring something new? What about clinicopathological features, survival, grade, stage...

It is a bit difficult to understand the methylation analysis. It seems that the heatmaps in Fig4 the authors plot the Beta-values for DNA methylation. Beta-values are usually [-1, 1]. The values in this plot seem different. It is in addition not described how the enrichment analysis is performed afterward. The authors could make better use of their differential methylation analysis and assess whether differentially methylated CpGs are in the binding site of specific transcription factor which may explain the biology between Claudin low in general or CLs subtypes.

Also, from the same Figure 4 the authors manage to cluster apparently a lot more Claudin-low tumors (x-axis). Where are these samples coming from? TCGA? It seems strange to me that there are so many CL in TCGA. I would not expect more than 50-60 CL in TCGA according to the CL calling.

In Figure 5, why using the median of the score this analysis would be much more powerful and convincing if the unsupervised clustering was produced at the individual sample level, as the authors conduct ssGSEA, this should be possible.

From Fig6G and summary Fig7, it seems clear that CL2 and CL3 are very related to Luminal and Basal-like subtypes. An arrow with EMT is then built. Is the EMT observed by the authors a difference in stromal/immune composition of these tumors? Increased Cancer associated Fibroblast in CL2 and CL3 could 'produce' these EMT related enrichment in pathways. How can the author address this?

Finally see: <https://www.biorxiv.org/content/10.1101/756411v1> and maybe comment not necessary in the text but at least for reviewers and editors how the results of this analysis is comparing to that reference.

Claudin-low is not a very well-defined subtype in breast cancer, as the author observe early in their analyses tumor purity is low. Further trying to subdivide a poorly characterized subtype in three subgroups is risky, especially when subdividing only a fraction of this poorly characterized subtype. In addition, the stromal/immune component of claudin low seems to be important as most often the tumor purity <50%. Therefore, trying to draw conclusions from the bulk molecular data of these tumor is also very risky especially when this stromal/immune component is ignored in the analysis. It would be also important to justify the temporal model described in the discussion and the last figure of the paper. What in the present analyses allows the authors to built this time succession of events? This will be very useful for the reader.

Reviewer #2:

Remarks to the Author:

In this manuscript, the authors examine claudin-low tumors, an aggressive subtype defined by low expression of cell junction components and associated with mesenchymal and stem features in the METABRIC dataset. Because gene expression for claudin-low tumors are similar to normal stroma, the researchers used allele specific copy number analysis (ASCAT) to estimate purity of tumor cell fractions using a stringent threshold. They characterized 42 low-claudin tumors by gene expression and copy number alterations and show a diversity of clinical subtypes, integrative cluster membership and fraction of genome altered (FGA). The authors employed a clever method of analyzing B allele frequencies and log-R ratios of SNPs located in genomic regions of deletion, to demonstrate that claudin-low tumors with a low FGA scores are likely not due to normal contamination but rather tumors with an absence of gross chromosomal instability. Using FGA they divided up the low-claudin subtype into three additional subtypes (CL1, CL2 and CL3). CL1, defined by a low FGA (< 10%), was mostly composed of ER-negative tumors that were all stratified into integrative cluster 4, CL2 was mostly composed of ER-positive tumors mainly stratified in two luminal-related clusters and CL3 displayed a high FGA (> 30%) classified into the genomically unstable cluster 10. The authors then demonstrate each CL subtype displays unique gene expression, pathway enrichment and methylation patterns. These data with correlations to normal mammary cell lineages suggests different cell origins for the various claudin-low subtypes and explain the variety of clinical subtypes in claudin-low tumors. The researchers also identify a potential dependency on MAPK signaling in claudin-low tumors, with cell lines more sensitive to MEK inhibitors, uncovering a potential future therapeutic option.

Specific comments

1) Unstratified claudin-low tumors are associated with worse overall survival, likely due to the enrichment of TNBC tumors. Are there differences in survival when sub-stratified into CL1, CL2, CL3. To achieve enough statistical power, METABRIC and TCGA may need to be combined in the survival analysis.

2) Given that Claudin-low tumors exhibit marked immune and stromal cell infiltration, were there any differences in immune cell composition in CL1, CL2 or CL3. The authors should consider using gene expression data estimations (ESTIMATE) and pathological evaluation of H&E sections, which

available for at least the TCGA (<https://cancer.digitalarchive.org/>) dataset. In addition, the authors should consider estimating immune cell composition with cibersort or xCell.

3) After purity threshold, were the 1270 remaining tumors re-subtyped for PAM50 with geneFu? Subtype calls for the same samples have the potential to shift greatly when subsets are analyzed.

4) Was there any enrichment in unique histological subtypes among the claudin-low subtypes?

Initial version	Revised Version
Main Figures 1 to 7	Main Figures 1 to 7
Supplementary Figure 1	Supplementary Figure 1
	Supplementary Figure 2 (new)
Supplementary Figure 2	Supplementary Figure 3
	Supplementary Figure 4 (new)
Supplementary Figure 3	Supplementary Figure 5
Supplementary Figure 4	Supplementary Figure 6
Supplementary Figure 5	Supplementary Figure 7
Supplementary Figure 6	Supplementary Figure 8
	Supplementary Figure 9 (new)
Supplementary Figure 7	Supplementary Figure 10
	Supplementary Figure 11 (new)

Reviewers' comments:

Reviewer #1 (Remarks to the Author):

1.1. *“The work of Pommier et al., describes a multiomics approach to analyze a rare/elusive subtype of breast cancer: claudin-low and to infer the cell of origin of this molecular class.*

In performing a comparison to non-Claudin-low tumors the authors, exclude the Claudin-low tumors with ‘outlier’ tumor purity very early in their analyses. They just choose to analyze the remaining 45 claudin-low tumors with sufficiently high tumor purity. Given that tumor subtypes are increasingly characterized by their microenvironment, this first step in the analysis is discussable.”

We agree with the comment made by Reviewer #1 that claudin-low tumors are generally considered as highly infiltrated by non-tumor cells present in the microenvironment. Considering this issue, it is important to stress that the main objective of our study (as highlighted in the introduction of the manuscript) was not to generally characterize claudin-low tumors, but rather to decipher the genomic architecture of these malignancies as a way to gain insight into their developmental origin. Hence, a purity-based selection of tumor samples was an absolute necessity (as also underlined by Reviewer #2) to unequivocally demonstrate the existence of three distinct claudin-low subgroups, and more specifically the existence of claudin-low tumors exhibiting a low level of chromosomal instability. The latter notion is of critical importance because samples with a paucity of genomic aberrations are generally considered as tumors with massive contamination. Of note, as now mentioned in the revised version of the manuscript, while most of our analyses were indeed performed on tumors selected for their high tumor purity, the 3 subgroups of claudin-low tumors are also found when studying the whole

cohort of claudin-low tumors (as illustrated in a new supplementary figure, referred as **Supplementary Figure 4**), further substantiating our data (see point **#1.2.**).

1.2. *“First, it means that the authors choose to analyze only 30% of the Claudin-low; should one not analyze all of them? Claudin-low represent approx. 10% of breast cancer cases, choosing to focus on only 30% of these 10% brings us to a very little proportion of breast cancer cases.”*

We agree with the reviewer that the number of samples analyzed following the purity-based selection was relatively low, though as discussed in **#1.1.**, this selection was an absolute necessity to decipher the genomic architecture of claudin-low tumors, the main objective of our study. To verify that the purity-selected tumor cohort was representative of the whole cohort (without purity selection), we validated that repartition of TNBCs within the molecular subtypes was not biased by this selection, as illustrated in **Supplementary Figure 1c-d** of both the initial and the revised version of the manuscript. In order to further address Reviewer #1's comment, we have now included a similar set of analyses as those performed in **Figure 2** but without applying tumor purity selection, illustrated in a new supplementary figure, referred as **Supplementary Figure 4** in the revised manuscript. Similarly to **Figure 2**, the new analysis presented in **Supplementary Figure 4** shows that claudin-low tumors form a genomically and molecularly heterogeneous subgroup, and present a trimodal distribution based on their FGA, thus validating the existence and the nature of the three claudin-low subgroups regardless of tumor purity selection.

1.3. *“While the authors further manage to subdivide the 45 cases further analyzed into three subgroups, one can question the relevance of such results when they are not verified in other cohorts than the METABRIC.”*

Contrary to Reviewer #1's statement, the results obtained in METABRIC cohort have been validated in both TCGA and CCLE cohorts, confirming the existence of the three claudin-low subgroups, as presented in **Supplementary Figure 6e-f** of the revised version of the manuscript (previously referred as **Supplementary Figure 4e-f** in the initial version of the manuscript).

1.4. *“Second it appears that, in general Claudin-low have very low tumor-percent. Have the authors checked whether within the discovered here subgroups of Claudin-low do not differ tumor percent? by this I mean: do Claudin-low in IC3 have same tumor percent than non-claudin low in IC3, same for IC4 and IC10?”*

If we correctly understand, there are two different questions raised by Reviewer #1.

First, is the tumor purity similar within the three claudin-low subgroups? This was shown in **Figure 5** of the manuscript (purity row in heatmap annotation), in which we conducted an unsupervised clustering analysis based on the ASCAT median purity score of each subtype. CL1 presents a higher purity score compared to CL2 and CL3 subgroups in METABRIC cohort, and a slightly lower purity score compared to CL2 and CL3 subgroups in TCGA cohort. Nevertheless, this difference in the purity score does not impact the clustering of the three subgroups.

The Reviewer #1's second question is related to the comparison of tumor purity between claudin-low and non-claudin-low tumors within the integrative clusters, previously defined by Curtis and colleagues (Curtis C. *et al.*, Nature 2012; Dawson S.-J. *et al.*, EMBO 2013). We have now addressed this issue, by performing the analysis requested and found that claudin-low subgroups do not differ statistically in purity score from non-claudin-low in all the three integrative clusters 3, 4 and 10. Data are shown in a new supplementary figure, referred as **Supplementary Figure 2** in the revised manuscript.

1.5. *“While the authors use all non-claudin low tumors to exclude the Claudin low with the lowest tumor percent, the rest of the 30% Claudin-low analyzed (the one with decent purity) are not called claudin-low because of a lower tumor percent according to the main subtype they ‘derive’ from? In other words: from the literature appears that the definition of claudin low subtype is intermingled with the microenvironment, so excluding it by setting filter on tumor purity the authors risk to produce skewed results.”*

We are not sure to understand Reviewer #1's concerns and expectations considering his two consecutive statements, claiming that our claudin-low selection is both (1) not stringent enough, suspecting that claudin-low are “called claudin-low because of a lower tumor percent”; and (2) too stringent, excluding the microenvironment by setting filter on tumor purity “risks to produce skewed results”.

Considering the first issue, it is noticeable that CL2 and CL3 display a statistically different purity from their non-claudin-low counterparts (CL2 vs luminal, $p=0.03$; CL3 vs basal, $p=0.04$), due to their intrinsic

feature of infiltrated tumors. Nevertheless, as illustrated in **Supplementary Figure 1** of the manuscript, we identified luminal and basal tumor samples with a low degree of purity, indicating that the attribution of a tumor sample to a molecular subtype is not prejudiced by a low tumor purity.

Considering the second issue, as explained above, the main objective of our study is to decipher the genomic architecture of these malignancies rather than to generally characterize claudin-low tumors and their microenvironment. Nevertheless, we fully agree with both Reviewers #1 and #2 that we cannot overlook the microenvironment, as it might influence the development of the tumors. Therefore, we have assessed the microenvironment composition through the analysis of gene expression *via* signature-based deconvolution method. Results are presented in a new supplementary figure, referred as **Supplementary Figure 9** in the revised manuscript. This analysis confirmed that, as previously reported (*Prat A. et al., Breast Cancer Research 2010; Dias K. et al., PLoS One 2017*), claudin-low tumors are among the breast malignancies that are highly infiltrated by immune cells (**Supplementary Fig. 9a-d**). Nevertheless, unsupervised clustering of microenvironment signatures (immune and non-immune cell subsets) did not allow to discriminate the different claudin-low subgroups (**Supplementary Fig. 9e-h**).

Overall, we agree that the purity selection strategy chosen for this study restricts the analysis of the full spectrum of claudin-low subgroup characteristics, but nonetheless it was the indispensable method to emphasize their unique genomic architecture. In particular, this analysis led to the demonstration of the existence of claudin-low tumors with a very low FGA. Moreover, we show that, while most of our analyses were indeed performed on tumors selected for their high tumor purity, the 3 subgroups of claudin-low tumors are also found when studying the whole cohort of claudin-low tumors (**Supplementary Fig. 4**).

1.6. *“Concerning the GSEA and the ssGSEA, would the authors have obtained the same results by comparing normal breast tissue, versus luminal versus basal tumors?”*

Reviewer #1 may suggest here that the CL1 subgroup could be normal breast tissue, and CL2 and CL3 could be luminal and basal tumors highly contaminated by non-tumor cells, calling into question the existence of claudin-low tumors as a true molecular subtype. Considering the CL1 subgroup, we have addressed this critical issue in **Figure 1b** of the manuscript, a pivotal figure panel in the paper acknowledged by Reviewer #2. In this figure, we identify regions of single copy allelic loss displaying a clear separation between BAF values and the presence of somatic mutations with a variant allele frequency higher than 0.4. Altogether, these results unequivocally demonstrate that ‘CNA devoid’

claudin-low tumors are not 'artefacts' due to contamination by normal cells but are genuine tumors that have developed in absence of periods of gross chromosomal instability.

1.7. *"Here, a more detail comparison with known subtype Luminal / Basal is warranted. The independence and originality of the CL subtypes is not clear in terms of pathway enrichment in Figure 3."*

We have already addressed this issue (**Supplementary Figure 3** of the initial version of the manuscript, now referred as **Supplementary Figure 5** in the revised version).

1.8. *"The authors chose the Gaussian mixture model of the FGA as the best way to find three distinct subgroups for Claudin-low. What is the justification for that? Other unbiased approaches could have been used such as unsupervised clustering of gene expression. Is the subdivision of the Claudin-low by PAM50 or integrative clusters a sufficient and relevant subdivision?"*

Although several methods of subdivisions are possible and mainly based on gene expression, the main objective of our study is to decipher the genomic architecture of these malignancies. This led us to analyze and subdivide claudin-low tumors according to their FGA level rather than their gene expression profile. Nevertheless, as illustrated in **Figure 2e-f** of the manuscript, the integrative cluster and PAM50 analyses generate similar but not entirely overlapping claudin-low subgroups compared to the FGA-based subdivision of claudin-low.

1.9. *"Are the Claudin-low subtypes observed by the authors clinically-relevant or does it bring something new? What about clinicopathological features, survival, grade, stage..."*

We thank Reviewer #1 for his useful comment. As also suggested by Reviewer #2, we have analyzed clinicopathological features of claudin-low and non-claudin-low tumors from METABRIC and TCGA combined cohorts. Data are shown in a new supplementary figure, referred as **Supplementary Figure 11** in the revised manuscript.

1.10. *“It is a bit difficult to understand the methylation analysis. It seems that the heatmaps in Fig4 the authors plot the Beta-values for DNA methylation. Beta-values are usually [-1, 1]. The values in this plot seem different. It is in addition not described how the enrichment analysis is performed afterward.”*

We are surprised by this comment since the detailed methylation and enrichment analysis strategy workflow was presented in **Supplementary Figure 5** in the initial version of the manuscript (now referred as **Supplementary Figure 7** in the revised version). The methylation levels, illustrated in **Figure 4** of the manuscript, are shown as scaled beta-value for a better graphical visualization. We fully agree that the annotation of the heatmaps in **Figure 4** could be misleading and we have annotated them appropriately in the **Figure 4** and the corresponding legend in the revised version of the manuscript (as in Jung et al., *Nature Communications* 2019).

1.11. *“The authors could make better use of their differential methylation analysis and assess whether differentially methylated CpGs are in the binding site of specific transcription factor which may explain the biology between Claudin low in general or CLs subtypes.”*

The question concerning the role of methylation on the binding of specific transcription factors and how it affects the biology of claudin-low subtypes is interesting, though we believe out of the scope of our current manuscript, and would require a specific investigation. Indeed, the choice we made was to perform a functional methylation analysis by selecting the genes for which methylation level was negatively correlated with gene expression. Thus, the methylation data used were pre-processed at the gene level and not at the probe level and therefore did not allow for a comprehensive interrogation of the methylated CpGs located in the binding sites of specific transcription factors targeting these genes.

Overall, we strongly believe that the analysis reported in the manuscript unequivocally demonstrates the distinct methylation profiles of claudin-low subgroups, correlated with gene expression.

1.12. *“Also, from the same Figure 4 the authors manage to cluster apparently a lot more Claudin-low tumors (x-axis). Where are these samples coming from? TCGA? It seems strange to me that here are so many CL in TCGA. I would not expect more than 50-60 CL in TCGA according to the CL calling.”*

Indeed, the tumor samples included in the analysis of **Figure 4** are from TCGA, as explained in the original figure legend. The analysis was conducted within the whole cohort of claudin-low breast

tumors (n=139), without applying tumor purity selection to ensure a sufficient statistical power when comparing within the claudin-low subgroups.

1.13. *“In Figure 5, why using the median of the score this analysis would be much more powerful and convincing if the unsupervised clustering was produced at the individual sample level, as the authors conduct ssGSEA, this should be possible.”*

We chose to perform the geneset expression analysis, displayed in **Figure 5** of the manuscript, with the median of each breast tumor subgroup to give an equal weight to each subtype regardless of their sample size (e.g.: 529 Luminal B versus 14 CL1 tumors). We then validated the identified pathways at the individual tumor sample level to illustrate intra-subtype heterogeneity, as shown in **Figure 6** and **Supplementary Figure 10** of the revised manuscript (previously referred as **Supplementary Figure 7** in the initial version of the manuscript).

1.14. *“From Fig6G and summary Fig7, it seems clear that CL2 and CL3 are very related to Luminal and Basal-like subtypes. An arrow with EMT is then built. Is the EMT observed by the authors a difference in stromal/immune composition of these tumors?”*

Since this comment is also addressed in the concluding remarks of Reviewer #1, we will provide a detailed explanation below (see **#1.17.**).

1.15. *“Increased Cancer associated Fibroblast in CL2 and CL3 could 'produce' these EMT related enrichment in pathways. How can the author address this?”*

As indicated in **#1.5**, we have assessed the microenvironment composition through the analysis of gene expression by using a signature-based deconvolution method. Results are shown in a new supplementary figure, referred as **Supplementary Figure 9** in the revised manuscript. The analysis did not highlight any enrichment in cancer-associated fibroblast signature in CL2 or CL3, as compared to luminal and basal tumors, respectively (**Supplementary Figure 9a-d**).

1.16. “Finally see: <https://www.biorxiv.org/content/10.1101/756411v1> and maybe comment not necessary in the text but at least for reviewers and editors how the results of this analysis is comparing to that reference.”

As mentioned in the cover-letter joint to our initial submission, the manuscript entitled “*Re-definition of claudin-low as a breast cancer phenotype*” by Fougner C *et al.*, available in BioRxiv (doi: <https://doi.org/10.1101/756411>) came to our attention as their data were highly complementary to our own. We strongly believe that our data strengthen their conclusions but also provide additional and original results with important implications on the genetic history of breast tumorigenesis.

Although based on different bioinformatics approaches and public databases (METABRIC and TCGA), data and conclusions of our current study are perfectly complementary with the results reported by Fougner and colleagues, strengthening the following take-home messages:

- the actual classifier classically used for breast tumor molecular classification (“9 cell-line classifier” from Prat *et al.*, *Breast Cancer Research* 2010) is inappropriate to accurately analyze claudin-low breast tumors, without considering normal (non-tumor) cell contamination (Fougner *et al.*: Figure 4 & Figure 5; Pommier *et al.*: **Supplementary Figure 1**).
- claudin-low breast tumors are a highly heterogeneous subtype of malignancies (Fougner *et al.*: Figure 1; Pommier *et al.*: **Figure 1a, Figure 2e-f & Supplementary Figure 1b,d**).
- claudin-low breast tumor subtype comprises distinct subgroups, resembling the full spectrum of non-claudin-low breast tumors at the levels of:
 - molecular classification (Fougner *et al.*: Figure 1a-b,e & Figure 4a,d; Pommier *et al.*: **Figure 1a, Figure 2e-f & Supplementary Figure 1b,d**),
 - mutation spectrum (Fougner *et al.*: Figure 1c,f-g & Figure 4b; Pommier *et al.*: **Figure 6d** & data not shown),
 - genomic alterations (copy number alterations (CNA)) (Fougner *et al.*: Figure 1d,h-i & Figure 4c; Pommier *et al.*: **Figure 2a-d & Supplementary Figure 6d-f**).

Fougner *et al.* on the one hand propose a new classification method for claudin-low analysis relying on the use of a 19-gene core list (Fougner *et al.*: Figure 3). We, on the other hand, investigated the mechanisms underlying the development of claudin-low malignancies explaining the diversity observed among this subtype. To avoid a potential bias due to the contamination of tumor cells by non-tumor cells, we selected the purest tumors for all analyses (Pommier *et al.*: **Supplementary Figure 1**). Using CNA and mutation analyses, we unequivocally demonstrate the existence of claudin-low tumors with very low FGA as full-blown tumors exhibiting a paucity of genomic aberrations, dismissing the prevailing hypothesis that this “CNA devoid” subgroup might actually be composed of samples

with a high level of infiltration of normal cells (Pommier *et al.*: **Figure 1b-c & Supplementary Figure 3**). This irrefutable demonstration allowed us to extend our investigation of claudin-low diversity through a comprehensive multi-omics analysis including:

- genomic alteration (CNA and mutations) analysis (Pommier *et al.*: **Figure 1b-c, Figure 2a-d, Figure 6d, Supplementary Figure 1a,c, Supplementary Figure 3 & Supplementary Figure 6d-f**),
- molecular classification analysis (Pommier *et al.*: **Figure 1a, Figure 2e-f & Supplementary Figure 1b,d**),
- differential gene expression analysis (Pommier *et al.*: **Figure 3a & Supplementary Figure 5**),
- normal mammary differentiation signature comparison (Pommier *et al.*: **Figure 3b-c**),
- differential methylation analysis (Pommier *et al.*: **Figure 4 & Supplementary Figure 7**),
- stemness and EMT marker expression analysis (Pommier *et al.*: **Supplementary Figure 8**),
- biological pathway analysis at transcriptional (Pommier *et al.*: **Figure 5, Supplementary Figure 6g, Figure 6a,c,e,g & Supplementary Figure 10a,c,e**) and proteomic (Pommier *et al.*: **Supplementary Figure 10b,d,f & Figure 6b,f,g**) levels,
- drug sensitivity (Pommier *et al.*: **Figure 6h**).

1.17. *“Claudin-low is not a very well-defined subtype in breast cancer, as the author observe early in their analyses tumor purity is low. Further trying to subdivide a poorly characterized subtype in three subgroups is risky, especially when subdividing only a fraction of this poorly characterized subtype. In addition, the stromal/immune component of claudin low seems to be important as most often the tumor purity <50%. Therefore, trying to draw conclusions from the bulk molecular data of these tumor is also very risky especially when this stromal/immune component is ignored in the analysis. It would be also important to justify the temporal model described in the discussion and the last figure of the paper. What in the present analyses allows the authors to build this time succession of events? This will be very useful for the reader.”*

Considering the first part of the Reviewer #1's concluding remark, we would like to emphasize again that the very reason of our study was to characterize the genomic architectures of claudin-low tumors as a way to gain insight into their developmental origin. It was therefore an absolute necessity to focus our analysis on purity-based selected tumors. We strongly believe that, the present characterization of the 3 subgroups of claudin-low tumors, associated with the first demonstration of the existence of claudin-low tumors with very low FGA, further supports the rationality of our approach.

As for the second part of this comment, the illustration presented in **Figure 7** of the initial version of the manuscript is not a demonstration but a model, as indicated in the discussion of the manuscript.

This model is supported by the observations made on purity-selected claudin-low tumors from this manuscript, and is highly coherent with previous data published in *Morel et al., Nature Medicine 2017*.

Reviewer #2 (Remarks to the Author):

2.1. *“Unstratified claudin-low tumors are associated with worse overall survival, likely due to the enrichment of TNBC tumors. Are there differences in survival when sub-stratified into CL1, CL2, CL3. To achieve enough statistical power, METABRIC and TCGA may need to be combined in the survival analysis.”*

We thank Reviewer #2 for his very interesting comment. This issue was also addressed by Reviewer #1 (see paragraph #1.9). As astutely suggested by Reviewer #2, we have thus performed a survival analysis on claudin-low tumors from METABRIC and TCGA combined cohorts after tumor purity selection. Data are shown in a new supplementary figure, referred as **Supplementary Figure 11** in the revised version of the manuscript.

2.2. *“Given that Claudin-low tumors exhibit marked immune and stromal cell infiltration, were there any differences in immune cell composition in CL1, CL2 or CL3. The authors should consider using gene expression data estimations (ESTIMATE) and pathological evaluation of H&E sections, which available for at least the TCGA (<https://cancer.digitalslidearchive.org/>) dataset. In addition, the authors should consider estimating immune cell composition with ciphersort or xCell.”*

We have performed the analysis of the microenvironment composition as suggested by Reviewer #2, and presented the results in a new supplementary figure, referred as **Supplementary Figure 9** in the revised manuscript. Of note, according to our results described in **Supplementary Figure 1**, we used the ASCAT (allele-specific copy number analysis of tumors) copy number-based tumor purity estimation method to assess tumor cell fraction rather than using the ESTIMATE gene-expression based method, which is less appropriate when analyzing tumors with mesenchymal features. Microenvironment (immune and non-immune) cell composition was estimated through the analysis of gene expression *via* signature-based deconvolution method using two different algorithms (MCPcounter, Becht E. *et al.*, *Genome Biology* 2016; xCell, Aran D. *et al.*, *Genome Biology* 2017) in both METABRIC and TCGA cohorts. This analysis confirmed that, as previously reported (*Prat A. et al., Breast*

Cancer Research 2010; Dias K. et al., PLoS One 2017), claudin-low tumors are highly infiltrated by immune cells (**Supplementary Fig. 9a-d**). Nevertheless, unsupervised clustering of microenvironment signatures (immune and non-immune cell subsets) did not allow to discriminate the different claudin-low subgroups (**Supplementary Fig. 9e-h**). Similar results were obtained regardless of the cohort analyzed (METABRIC: **Supplementary Fig. 9a-b,e-f**; TCGA: **Supplementary Fig. 9c-d,g-h**) and the algorithms used (MCP counter: **Supplementary Fig. 9a,e,c,g**; xCell: **Supplementary Fig. 9b,d,f,h**).

Moreover, as requested by Reviewer #2, we have analyzed the data from TCGA of pathological evaluation of breast tumor tissue slides by molecular subtypes. As illustrated below, this analysis did not discriminate the three claudin-low subgroups and we decided not to include this figure in the manuscript.

Additional Figure: Pathological evaluation of TCGA breast tumor tissue slides by molecular subtypes
a- Proportion of different cell types (mean +/- sd), **b-** percentage of lymphocyte cell infiltration, **c-** percentage of monocyte cell infiltration and **d-** percentage of neutrophil cell infiltration.

2.3. "After purity threshold, were the 1270 remaining tumors re-subtyped for PAM50 with geneFu? Subtype calls for the same samples have the potential to shift greatly when subsets are analyzed."

The reliability of the expression-based classifiers was taken into account in our study. Indeed, to establish the subtyping of breast tumor samples with the most optimal robustness, we used 5 different classifiers (PAM50, AIMS, SCMGENE, SSP2006 and SMOD2) rather than using only PAM50, as mentioned in the *Methods* section of the manuscript. Re-subtyping was not performed after purity

selection in order to preserve the most extensive and unbiased cohort for molecular subtype calling. Nevertheless, we have checked the robustness of the subtyping after purity selection by analyzing the molecular subtype attribution with and without purity selection and checking for accuracy, as presented in the accuracy table below.

		Before purity threshold				
		Basal	Her2	LumA	LumB	Normal
After purity threshold	Basal	140	1	0	1	0
	Her2	3	86	0	12	0
	LumA	0	1	409	42	0
	LumB	0	14	57	485	0
	Normal	1	1	5	2	6

accuracy: 88.94155 %

Additional Table: Accuracy between breast cancer molecular subtype assignment before and after tumor purity threshold.

2.4. "Was there any enrichment in unique histological subtypes among the claudin-low subtypes?"

As also suggested by Reviewer #1 in #1.9., we have analyzed clinicopathological features (including tumor histological subtype, and also tumor stage, lymph node involvement and patient age, overall survival and disease-specific) in claudin-low and non-claudin-low tumors from METABRIC and TCGA combined cohorts after tumor purity selection. These data are now shown in a new supplementary figure, referred as **Supplementary Figure 11** in the revised manuscript.

To conclude, we greatly appreciated the thoughtful comments of the reviewers and we believe that we have addressed each of them. We hope that the proposed modifications substantially improve our manuscript making it a valuable publication for the journal.

We are looking forward to hearing from you and thank you for the opportunity to resubmit our work for publication in *Nature Communications*.

Yours sincerely,

Pr. Alain Puisieux

Reviewers' Comments:

Reviewer #1:

Remarks to the Author:

Many thanks for the review.

the Authors have replied sufficiently to my comments. They can prove that, contrary to my suspicion, the observed claudin low clusters are independent on the immune infiltration, yet reflected in the classification when also tumors with low purity are included. What one continue With to increase the novelty of this publication is to highlight the clinical importance of that finding, the actionable targets and pathways. All presented data represent re-analysis of the two larges available datasets on breast cancer. No new data are presented, or no experimental part is included that might confirm the suggested paths of tumor evolution, directly from normal or through a basal or luminal phenotype.

Clearly, the PAM50 classification, or the integrative clusrering are all categories imposed on a continuum. SUBstructures in them can be found (the Lehman classification of the basal tumors, etc) and some may be presented by the CL1-3 clutsters presented here. The Authors carefully highlight the foundings that corroborate the paper on Faugner et al, but need to highlight the novel, different and clinically useful findings in their paper.

Survuvall analysis is performed, but I could not see the p-values or a more profound discussion of the results. From Suppl figure 11 it appears that the CL clustering identifies a Group With very good prognosis (CL1) and one with very poor prognosis, and the discrimination seems superior to that of PAM50. If this is true, this may be a potential punchline, i.e. that with fewer markers one can identify these Extreme Groups of patients who may and who may not need additional treatment. This would be of great clinical utility.

Reviewer #2:

Remarks to the Author:

The authors have sufficiently addressed all of my concerns and significantly improved the manuscript in this revision.

Reviewers' comments:

Reviewer #1 (Remarks to the Author):

1.1. *"Many thanks for the review.*

The authors have replied sufficiently to my comments. They can prove that, contrary to my suspicion,

the observed claudin low clusters are independent on the immune infiltration, yet reflected in the classification when also tumors with low purity are included. What one continue with to increase the novelty of this publication is to highlight the clinical importance of that finding, the actionable targets and pathways. All presented data represent re-analysis of the two large available datasets on breast cancer. No new data are presented, or no experimental part is included that might confirm the suggested paths of tumor evolution, directly from normal or through a basal or luminal phenotype. Clearly, the PAM50 classification, or the integrative clustering are all categories imposed on a continuum. Substructures in them can be found (the Lehman classification of the basal tumors, etc) and some may be presented by the CL1-3 clusters presented here. The authors carefully highlight the findings that corroborate the paper on Fougner et al, but need to highlight the novel, different and clinically useful findings in their paper."

We appreciate the Reviewer #1 remarks. Here, we have characterized the intrinsic diversity within the claudin-low breast cancers by demonstrating the existence of three main molecular subgroups. Consistent with the data from the study by Fougner *et al.*, published during the revision process of our manuscript, we have shown that these subgroups are associated with distinct survival outcomes. Moreover, our study has delved into the underlying reasons of the observed heterogeneity, revealing that claudin-low tumors exhibit different developmental origins. Furthermore, the identification of the recurrent activation of the RAS-MAPK pathway in claudin-low breast cancers may have significant clinical implications in light of the previously described resistance of EMT-related tumors to a variety of anti-cancer therapies.

We have addressed the reviewer's comments by highlighting the clinical implications of our findings

in the discussion section of the revised manuscript.

1.2. *“Survival analysis is performed, but I could not see the p-values or a more profound discussion of the results. From Suppl figure 11 it appears that the CL clustering identifies a Group With very good prognosis (CL1) and one with very poor prognosis, and the discrimination seems superior to that of PAM50. If this is true, this may be a potential punchline, i.e. that with fewer markers one can identify these Extreme Groups of patients who may and who may not need additional treatment. This would be of great clinical utility.”*

We thank the Reviewer #1 for this comment. We have updated the Supplementary Figure 11 with the significance of the survival analysis, and the results were addressed in the discussion section of the revised manuscript.

Reviewer #2 (Remarks to the Author):

2.1. *“The authors have sufficiently addressed all of my concerns and significantly improved the manuscript in this revision.”*

We thank the Reviewer #2 for the valuable comments.